# Antimicrobial Activity of EPA and DHA against Oral Pathogenic Bacteria Using an In Vitro Multi-Species Subgingival Biofilm Model

**DOI:** 10.3390/nu12092812

**Published:** 2020-09-14

**Authors:** Honorato Ribeiro-Vidal, María Carmen Sánchez, Andrea Alonso-Español, Elena Figuero, Maria José Ciudad, Luís Collado, David Herrera, Mariano Sanz

**Affiliations:** 1ETEP (Etiology and Therapy of Periodontal and Peri-implant Diseases) Research Group, University Complutense of Madrid, 28040 Madrid, Spain; honoribe@ucm.es (H.R.-V.); mariasan@ucm.es (M.C.S.); andalons@ucm.es (A.A.-E.); efigueruiz@odon.ucm.es (E.F.); davidher@odon.ucm.es (D.H.); 2Medicine Department, Faculty of Medicine, University Complutense of Madrid, 28040 Madrid, Spain; mjciudad@ucm.es (M.J.C.); lcollado@ucm.es (L.C.)

**Keywords:** omega-3 fatty acids, docosahexaenoic acid, eicosapentaenoic acid, oral biofilms, confocal laser microscopy, PUFAs, prevention, periodontal diseases

## Abstract

In search for natural products with antimicrobial properties for use in the prevention and treatment of periodontitis, the purpose of this investigation was to evaluate the antimicrobial activity of two omega-3 fatty acids, docosahexaenoic acid (DHA) and eicosapentaenoic acid (EPA), using an in vitro multi-species subgingival biofilm model including *Streptococcus oralis*, *Actinomyces naeslundii*, *Veillonella parvula*, *Fusobacterium nucleatum*, *Porphyromonas gingivalis*, and *Aggregatibacter actinomycetemcomitans*. The antimicrobial activities of EPA and DHA extracts (100 µM) and the respective controls were assessed on 72 h biofilms by their submersion onto discs for 60 s. Antimicrobial activity was evaluated by quantitative polymerase chain reaction (qPCR), confocal laser scanning microscopy (CLSM) and scanning electron microscopy (SEM). ANOVA with Bonferroni correction was used to evaluate the antimicrobial activity of each of the fatty acids. Both DHA and EPA significantly reduced (*p* < 0.001 in all cases) the bacterial strains used in this biofilm model. The results with CLSM were consistent with those reported with qPCR. Structural damage was evidenced by SEM in some of the observed bacteria. It was concluded that both DHA and EPA have significant antimicrobial activity against the six bacterial species included in this biofilm model.

## 1. Introduction

Periodontitis is a chronic inflammatory disease associated with dysbiotic subgingival biofilms and characterised by the progressive destruction of the tooth supporting apparatus. Its primary features include the loss of periodontal attachment to the tooth, manifested by clinical attachment loss, radiographic bone loss, and presence of periodontal pockets and gingival bleeding [1].

Although the majority of the microorganisms that colonise the oral cavity and dental surfaces are compatible with periodontal health [2], there are specific pathobionts that have shown pathogenicity by disrupting the host immune tolerance and causing a chronic unresolved inflammation in the periodontal tissues, leading to destructive changes in the connective and bone tissue metabolism [3,4]. In particular, *Porphyromonas gingivalis* has been identified as an example of keystone pathogen, with the capacity to augment the virulence of the entire microbial community through specific inter-bacterial interactions, a characteristic feature of the “biofilm quorum sensing” [5,6], and the expression of certain molecules acting as virulence factors, like proteolytic enzymes or other pro-inflammatory molecules, that will induce a dysbiosis state by modifying the biofilm towards a pro-inflammophilic environment, thus promoting a non-resolving chronic inflammatory host response, what is characteristic of the subgingival biofilm in periodontitis. It should also be taken in consideration that differences intrinsic to the host response of each individual might influence the establishment and progression of the disease [7,8].

The current strategies to prevent and treat periodontitis are based in the reduction of the subgingival *biofilm* below a threshold compatible with homeostasis and health [9]. These strategies are based on mechanical root instrumentation, either non-surgically [10] or surgically [11], with or without adjunctive therapies, such as the use of locally and/or systemically delivered antimicrobials [12,13]. The use of adjunctive systemic antibiotics, although demonstrating a significant additional effect [13], may also cause unwanted effects [13,14,15], mainly associated with the development of bacterial resistances [16], which may represent a threat to global public health [17]. The use of antiseptics can also cause side effects, such as irritation of the mucous membranes, tooth staining or accelerated formation of dental calculus [18]. To overcome these limitations, the search for natural products with antimicrobial properties has been fostered and investigated [18,19,20,21].

One of these strategies has been the study of long-chain polyunsaturated fatty acids (PUFAs), fish and fish oil derivatives. PUFAs have demonstrated antimicrobial activity, with a broad spectrum inhibitory effect against various Gram-positive and Gram-negative bacteria [22,23,24,25,26,27,28]. Eicosapentaenoic acid (EPA) and docosahexaenoic acid (DHA) have shown antibacterial activity against different oral bacterial pathogens, such as *Streptococcus mutans*, *Candida albicans*, *P. gingivalis*, *Fusobacterium nucleatum* and *Prevotella intermedia* [29]. Their mechanism of action seems to be through blocking essential bacterial processes at the level of the plasmatic membrane, such as the electron transport chain and the oxidative phosphorylation [30,31,32,33,34,35].

In addition to this antimicrobial effect, PUFAs are substrates for the cyclooxygenase and lipoxygenase pathways, actively promoting the lipoxygenase pathway, thus stimulating the synthesis of lipoxins involved in the resolution of inflammation, and blocking the cyclooxygenase pathway, thus inhibiting the secretion of prostaglandins, potent pro-inflammatory and bone resorting molecules [36]. This double potential activity (antimicrobial and anti-inflammatory) has increased attention to these natural compounds as possible adjunctive alternatives in the prevention and treatment of periodontitis. However, although there are reports on the antimicrobial activity of PUFAs using planktonic bacteria or monospecies biofilms [37], there are no reports using validated multispecies subgingival models which better resemble real conditions [38]. It was, therefore, the purpose of this investigation to to evaluate the antimicrobial activity of pure EPA and DHA against against oral pathogenic bacteria, using a validated multispecies in vitro biofilm model [38].

## 2. Materials and Methods

### 2.1. Omega 3 Fatty-Acids

The PUFAs independently evaluated in this investigation were EPA and DHA, obtained already solubilized in ethanol (Cerilliant^®^, Sigma-Aldrich, Barcelona, Spain).

### 2.2. Bacterial Strains and Culture Conditions

Reference strains of *Streptococcus oralis* CECT 907T, *Veillonella parvula* NCTC 11810, *Actinomyces naeslundii* ATCC 19039, *F. nucleatum* DMSZ 20482, *Aggregatibacter actinomycetemcomitans* DSMZ 8324, and *P. gingivalis* ATCC 33277 were used. These bacteria were grown on blood agar plates (Blood Agar Oxoid No 2; Oxoid, Basingstoke, UK), supplemented with 5% (*v*/*v*) sterile horse blood (Oxoid), 5.0 mg L−1 hemin (Sigma, St. Louis, MO, USA) and 1.0 mg L−1 menadione (Merck, Darmstadt, Germany) in anaerobic conditions (10% H2, 10% CO2, and balance N2) at 37 °C for 24–72 h.

### 2.3. Antibacterial Effect of EPA and DHA against Planktonic Bacteria

For determining which concentration of each of EPA and DHA was appropriate for the biofilm model assays, we undertook independent previous microtiter plate-based antibacterial assays for each of the studied fatty acids. In brief, pure cultures of the six selected bacterial strains were grown anaerobically in a protein rich medium containing brain–heart infusion (BHI) (Becton, Dickinson and Company, Franklin Lakes, NJ, USA) supplemented with 2.5 g L^−1^ mucin (Oxoid), 1.0 g L^−1^ yeast extract (Oxoid), 0.1 g L^−1^ cysteine (Sigma), 2.0 g L^−1^ sodium bicarbonate (Merck), 5.0 mg L^−1^ hemin (Sigma), 1.0 mg L^−1^ menadione (Merck), and 0.25% (*v*/*v*) glutamic acid (Sigma). At mid-exponential phase of bacterial growth (measured by spectrophotometry), bacteria were placed on a 96-well microtitre plates adding 200 μL of a mixture of each bacteria inoculum at a final concentration of 10^6^ colony-forming units (CFUs) mL^−1^, and EPA or DHA for a final concentration of 12.5, 25, 50, 100 and 200 μM. Plates had a set of controls: phosphate-buffered saline (PBS) was used as negative control; ethanol controls (adjusted to match the ethanol concentration present in each of the fatty acids (EtOH)); positive control (bacteria without any treatment). A measurement (optical density, O.D.595) as t = 0 absorbance was taken in a microtitre plate reader (Optic Ivymen System 2100-C; I.C.T.; La Rioja, Spain). The microplates were incubated for 48 h at 37 °C under anaerobic conditions, and absorbance was measured each 2 h, in order to determine the bacterial growth until reaching a stationary growth phase. Minimum inhibitory concentration (MIC) and minimum bactericidal concentration (MBC) values were calculated and confirmed by microbial plate counting on blood agar media. Accordingly, the lowest concentration of the DHA or EPA showing growth inhibition was considered as the MIC, whereas the lowest concentration of the DHA or EPA that showed zero growth in blood agar plates, after spot inoculation and incubation for 72 h, was recorded as the MBC. All experiments were performed in triplicate with appropriate controls.

### 2.4. Biofilm Development

A multispecies in vitro biofilm model was developed, as previously described by Sánchez and co-workers [38]. Briefly, pure cultures of each bacteria were anaerobically grown in a modified BHI liquid medium. Bacterial cultures were harvested at mid-exponential phase (measured by spectrophotometry), and a mixed bacteria suspension in modified BHI medium, containing 10^3^ CFU mL^−1^ for *S. oralis*, 10^5^ CFU mL^−1^ for *V. parvula* and *A. naeslundii*, and 10^6^ CFU mL^−1^ for *F. nucleatum*, *A. actinomycetemcomitans* and *P. gingivalis*, was prepared. Sterile calcium hydroxyapatite (HA) discs, of 7 mm of diameter and 1.8 mm (standard deviation, SD = 0.2) of thickness (Clarkson Chromatography Products, Williamsport, PA, USA), were coated with treated saliva for 4 h at 37 °C in sterile plastic tubes, and then placed in the wells of a 24-well tissue culture plate (Greiner Bio-one, Frickenhausen, Germany). Each well was inoculated with 1.5 mL of mixed bacteria inoculum and incubated in anaerobic conditions (10% H_2_, 10% CO_2_, and balance N_2_) at 37 °C for 72 h. At 37 °C for 72 h, the timepoint in which the biofilm model reach maturity, containing all bacterial species at an optimal concentration to carry out the assay [38,39,40]. The plates employed for assessing the sterility of the culture medium were used as controls.

### 2.5. Antimicrobial Activity on Biofilms

The antimicrobial activities of EPA and DHA extracts (100 µM) were assessed independently on 72 h biofilms by their submersion onto discs for 60 s. Phosphate-buffered saline (PBS) was used as a negative control; ethanol controls (at the same concentration of the one in commercial DHA or EPA) were used to rule out the bactericidal effect of the solvent (EtOH); and 0.2% chlorhexidine (Sigma-Aldrich) was used as positive control.

All the independent sets of experiments for each of the PUFAs were repeated three times on different days using fresh bacterial cultures with trios of biofilms for each analysis.

### 2.6. Microbiological Outcomes: Quantitative Polymerase Chain Reaction (qPCR) Analysis

After the application of the tested products, treated biofilms were sequentially rinsed in 2 mL of sterile PBS three times (immersion time per rinse, 10 s), disrupted by vortex for 2 min in 1 mL of PBS and treated with a 100 µM concentration of propidium monoazide (PMA) (Biotium Inc., Hayword, CA, USA) to discriminate between DNA from live and dead bacteria [41]. Following an incubation period of 10 min at 4 °C in the dark, the samples were subjected to light-exposure for 30 min, using a PMA-Lite LED Photolysis Device (Biotium Inc.). After PMA photo-induced DNA cross-linking, the cells were centrifuged at 12,000 rpm for 3 min prior to DNA isolation. To avoid any influence of the experimental process on bacterial viability, the same procedure (incubation at 4 °C and exposure to light source) but without the exposure to PMA, was used as a negative control.

Bacterial DNA was isolated from all biofilms using a commercial kit ATP Genomic DNA Mini Kit^®^ (ATP Biotech. Taipei, Taiwan), following manufacturer’s instructions, and the hydrolysis 5’nuclease probe assay qPCR method was used for detecting and quantifying the bacterial DNA. The qPCR amplification was performed following a protocol previously optimized by our research group, using primers and probes targeted against 16S *rRNA* gene (Life Technologies, Thermo Fisher Scientific, Carlsbad, CA, USA) [42]. Each DNA sample was analysed in duplicate. Quantification cycle (Cq) values, describing the PCR cycle number at which fluorescence rises above the baseline, were determined using the provided software package (LC 480 Software 1.5; Roche Diagnostic GmbH; Mannheim, Germany). Quantification of viable cells by qPCR was based on standard curves. The correlation between Cq values and CFU mL^−1^ was automatically generated by the software (LC 480 Software 1.5; Roche Diagnostic GmbH; Mannheim, Germany). All assays were run with a linear quantitative detection range established by the slope of 3.3–3.6 cycles/log decade, r^2^ > 0.997 and an efficiency range of 1.9–2.0.

### 2.7. Confocal Laser Scanning Microscopy (CLSM) Analysis

The CLSM analyses were performed at the Centre for Cytometry and Fluorescence Microscopy of the Complutense University of Madrid (UCM), Spain. Hydroxyapatite containing the grown biofilms were washed three times sequentially in 2 mL sterile PBS (10 s immersion per wash) to remove any remnants of the extracts and non-binding bacteria. Three separate, representative locations were selected on the HA discs covered with fully hydrated biofilms (based on the presence of columns or towers of bacterial communities, identified in the confocal field of vision) and analysed with non-invasive confocal microscopy using an Ix83 Olympus fixed-phase microscope coupled to an Olympus FV1200 confocal system (Olympus, Shinjuku, Tokyo, Japan). The specimens were stained with the LIVE/DEAD^®^ BacLight^TM^ Bacterial Viability Kit solution (Molecular Probes B.V., Leiden, The Netherlands) at room temperature. Fluorochrome at a ratio of 1:1 was used with a staining time of 9 ± 1 min in order to obtain the optimal fluorescence signal at the corresponding wavelengths (Syto9: 515–530 nm, propidium iodide (PI): >600 nm). The CLSM software was programmed to perform a series z of scans (xyz) 1 µm thick (16 bits, 2048 × 2048 pixels). The images were analysed using Olympus^®^ software (Olympus). Using the Fiji software (ImageJ Version 2.0.0-rc-65/1.52b, Open source image processing software), a live/dead analysis was performed in order to access the live/death ratio (green cells divided by the sum of green and red cells). Data were expressed as mean and SD.

### 2.8. Scanning Electron Microscopy (SEM) Analysis

The SEM analyses were performed at the National Centre of Electronic Microscopy (UCM, Madrid, Spain). Firstly, the samples were washed sequentially with 2 mL sterile PBS to remove non-binding bacteria on the HA disc and this process was repeated three times consecutively (10-s immersion per wash). After this, the samples were fixed with a solution of 4% paraformaldehyde (Panreac. Química, Barcelona, Spain) and 2.5% glutaraldehyde (Panreac. Química) for 4 h at 4 °C. Next, the samples were once again washed with PBS and sterile water (10 min immersion time per wash) and dehydrated through a graduated series of ethanol solutions (30, 50, 70, 80, 90 and 100%; 10-min immersion time for each series). Then, the samples were dried by critical points, coated with gold by sputtering and analysed using electron microscopy, using a JSM 6400 electron microscope to do so (JSM 6400; JEOL, Tokyo, Japan), with a backscattered electron detector and an image resolution of 25 kV.

### 2.9. Statistical Analyses

The primary outcome variable was the number of viable bacteria present in the biofilm, measured by qPCR, for each tested bacterial species: *S. oralis*, *A. naeslundii*, *V. parvula*, *A. actinomycetemcomitans*, *P. gingivalis* and *F. nucleatum*, expressed as CFU mL^−1^. Form the mean values of each group, the percentage of reduction was calculated for DHA or EPA, EtOH and CHX when compared to the negative control (PBS) value [43]. As a secondary outcome, the live/death ratio obtained through the CLSM analysis was compared among the groups.

To determine the data distribution, box plots, asymmetry coefficients and Shapiro–Wilk tests were used. Data were expressed as means and SD. Two ANOVA tests with post-hoc Bonferroni corrections were performed in order to independently compare DHA or EPA versus PBS, EtOH and CHX. Data analysis was carried out using a dedicated computer software (IBM SPSS Statistics 24.0; IBM Corporation, Armonk, NY, USA) and the results were considered statistically significant at *p* ≤ 0.05.

## 3. Results

### 3.1. Antibacterial Effect of DHA and EPA on Planktonic Bacteria

MICs and MBCs values against the six bacterial strains selected in planktonic state were determined for each of the fatty acids. In the case of DHA, the MICs were 50 μM for *S. oralis*, *A. naeslundii* and *V. parvula*; 100 µM for *F. nucleatum*; and 25 μM for *P. gingivalis* and *A. actinomycetemcomitans*. The MBCs were 100 µM for all the six bacterial species.

In the case of EPA, the MICs were 50 μM for *S. oralis* and *V. parvula,* and 100 µM for *A. naeslundii* and *F. nucleatum*, and 25 μM for *P. gingivalis* and *A. actinomycetemcomitans*. The MBCs were 100 µM for all six bacterial species. Based on these results, the 100 μM concentration was selected in the biofilm experiments with the two fatty acids.

### 3.2. Antibacterial Effect of the DHA Extract on Biofilm

#### 3.2.1. Analysis by qPCR

The results of the antimicrobial effect of DHA extracts, compared with the negative control (PBS), EtOH and the positive control (CHX), are depicted in Table 1 and expressed as average counts of viable bacteria recovered from the 72 h biofilms (viable CFU mL^−1^). Compared with the negative control (PBS), a statistically significant reduction (*p* < 0.001) in viable bacterial counts of the initial and early colonizers was observed. These reductions reached three orders of magnitude, amounting to 99.96% for *S. oralis*, 99.82% for *A. naeslundii* and 99.80% for *V. parvula* (Table 1). No statistically significant differences were found when comparing DHA with EtOH or CHX, although the magnitudes of reduction achieved by these controls were lower than with DHA (Table 2).

For the secondary coloniser *F. nucleatum*, statistically significant reductions (*p* < 0.001) were observed when comparing DHA to the negative control (99.92%). Differences were also statistically significant when comparing DHA with CHX (*p* < 0.002), with CHX showing a lesser effect compared with DHA. When comparing EtOH and CHX with the negative control, smaller differences were observed (Table 2).

For the periodontal pathogens *P. gingivalis* and *A. actinomycetemcomitans*, statistically significant reductions occurred after exposure with DHA, EtOH and CHX (*p* < 0.001, in all cases), when compared to the negative control. With DHA, these reductions were up to three orders of magnitude for *P. gingivalis* (99.92%) and for *A. actinomycetemcomitans* (99.90%), while after exposure to EtOH or CHX, the reductions were of a single order of magnitude. 

#### 3.2.2. CLSM Analysis

In control biofilms, after 72 h of incubation, CLSM images showed the entire surface of the HA discs covered by bacteria, with a live/death ratio of 0.74 (SD = 0.07). These biofilms showed the morphological characteristics of multi-species bacterial communities grouped in “towers” (Figure 1A). After 60 s of exposure to the DHA extracts, the reduction in cell viability was pronounced (Figure 1C), with a live/death ratio of 0.11 (SD = 0.08). This reduction also occurred in the biofilms exposed to EtOH (Figure 1B), although it was of a smaller magnitude (0.49, SD = 0.08). Similar reductions were also observed when exposed to 0.2% CHX (Figure 1D), with a live/death ratio of 0.37 (SD = 0.07).

#### 3.2.3. SEM Analysis

*Biofilms* were clearly identified by SEM on the HA discs after treatment with PBS, DHA, EtOH or CHX (Figure 2).

On the HA discs treated with PBS (Figure 2A), bacterial growth on the surface of the discs was evident, with bacterial cells forming chains (characteristic of *Streptococcus* and *Aggregatibacter* species), or as multicellular aggregates, with a structural organisation based primarily on cell-to-cell co-aggregation. The fusiform bacilli, characteristic of *F. nucleatum* species, were also recognizable, forming three-dimensional structures resembling sessile communities.

The analysis of the discs treated with DHA (Figure 2B) showed marked differences compared to the control discs, such as a lower bacterial density in the surface, and a high proportion of bacteria with evident structural damage. The discs treated with EtOH (Figure 2C) or CHX (Figure 2D) also showed a reduction in bacterial density, but it was not as marked as in the discs treated with DHA (Figure 2B).

### 3.3. Antibacterial Effect of the EPA Extract

#### 3.3.1. qPCR Analysis

The antimicrobial effects of the EPA extracts, compared to the negative control (PBS), EtOH and the positive control (CHX), are depicted in Table 3, expressed as average counts of viable bacteria recovered from the 72 h biofilms.

Compared to the negative control (PBS), statistically significant reductions in viable bacteria counts of *S. oralis* were seen after the exposure to EPA (*p* < 0.001), reaching a reduction of 97.16% in CFUs mL^−1^ (Table 3). Compared to EtOH (*p* = 0.144) and CHX 0.2% (*p* = 1.000), no statistically significant differences were observed (Table 4).

For *A. naeslundii* and *V. parvula*, statistically significant reductions (98.36% and 95.43%, respectively) occurred after exposure to EPA (*p* < 0.001). With EtOH and CHX, the bacterial counts were also significantly reduced, albeit to a lesser magnitude. For *V. parvula*, statistically significant differences were also found when comparing the effect of EPA with EtOH (*p* < 0.003). For *F. nucleatum*, statistically significant reductions (98.67%) occurred with the exposure to EPA (*p* < 0.001), with a similar lesser effect when EtOH and CHX 0.2% were applied. Furthermore, the effect of EPA versus CHX was statistically significant (*p* < 0.001). For *P. gingivalis* and *A. actinomycetemcomitans*, statistically significant reductions occurred after the exposure to EPA (97.51% and 91.36%, respectively). Similar significant reductions were observed with EtOH and CHX (*p* < 0.001), when compared with PBS. For *P. gingivalis*, statistically significant differences were found when comparing exposure to the EPA extracts versus EtOH (*p* = 0.002) and versus CHX (*p* = 0.014). For *A. actinomycetemcomitans*, no differences were found, either when comparing exposure to EPA versus EtOH (*p* = 0.676) or versus CHX (*p* = 0.423) (Table 4).

#### 3.3.2. CLSM Analysis

In control biofilms, after 72 h of incubation, CLSM images showed that the entire surface of the HA discs was covered by live bacteria, with a live/death ratio of 0.75 (SD = 0.08). Biofilms depicted the characteristic morphology of bacterial communities grouped in multi-species “towers” (Figure 3A). In contrast, after 60 s of exposure to the EPA extracts, the reduction in cell viability was very pronounced (Figure 3C) being the live/death ratio 0.15 (SD = 0.09). This reduction also occurred in the biofilms exposed to EtOH (Figure 3B), although it was of a smaller magnitude 0.53 (SD = 0.09). Similar reductions were observed when applying 0.2% CHX (Figure 3D) where the live/death ratio 0.39 (SD = 0.06). These results were consistent with the qPCR results.

#### 3.3.3. SEM Analysis

*Biofilm* formation was clearly observed on the HA discs at 72 h after treatment with PBS, EPA, EtOH or CHX (Figure 4).

On the HA discs treated with PBS (Figure 4A), there was evidence of bacterial growth on the surface of the discs with identification of bacterial cells forming chains (*Streptococcus* and *Aggregatibacter* species) and the characteristic fusiform bacilli (*F. nucleatum* species), forming three-dimensional sessile communities. The analysis of the discs treated with EPA (Figure 4B) showed a lower bacterial density in the surface, with the presence of bacteria with evident structural damage. The discs treated with EtOH (Figure 4C) or CHX (Figure 4D) also showed a reduction in bacterial density, but it was not as marked as in the discs treated with EPA (Figure 4B).

## 4. Discussion

The results of the present investigation provide evidence that the application of EPA or DHA extracts, on mature biofilms grown on HA discs, significantly reduced the bacterial counts and the cell viability in the six bacterial strains used in this in vitro multispecies biofilm model (*S. oralis*, *A. naeslundii*, *V. parvula*, *F. nucleatum*, *P. gingivalis* and *A. actinomycetemcomitans*).

These results are concordant with previous investigations that have shown an antimicrobial antiseptic effect of PUFAs extracts [33], mainly EPA and DHA extracts found naturally in many marine organisms [44,45]. In addition, and beyond their antimicrobial activity (confirmed in the present study), it should be highlighted that PUFAs have also demonstrated anti-inflammatory [25,26,27,28,44,45,46,47] and antioxidant [48] properties.

These previous investigations reporting antibacterial effects of EPA and DHA used single-bacteria biofilm models in planktonic growth [29,49]. Sun et al., in 2016, investigated the potential effects of EPA and DHA against periodontal bacteria as mono-species biofilms in planktonic state, demonstrating significant antimicrobial activity against *P. gingivalis* and *F. nucleatum* [37]. However, it is widely known that microorganisms have different properties when growing within multispecies biofilms, compared to their planktonic state, such as resistance to antimicrobial agents [50,51,52,53,54,55]. A number of studies indicate that the MIC of an organism may increase 2- to 1000-fold when growing within a biofilm [52], being at least 250 times greater than the MIC of the same species in a planktonic state [56]. In fact, the bacterial resistance to antimicrobials appears to be related to the maturation of the biofilm, with maximum resistance coinciding with the stationary growth phase of the biofilm [56,57]. For these reasons, an in vitro multi-species biofilm model was used in the present investigation. This model has been previously validated by Sanchez et al. and it included six species commonly present in subgingival biofilms (*S. oralis*, *A. naeslundii*, *V. parvula*, *F. nucleatum*, *P. gingivalis* and *A. actinomycetemcomitans*) [38]. In the present investigation, EPA and DHA were evaluated independently, compared with negative and positive controls, demonstrating that DHA extracts may have a superior effect, against these controls, although a direct comparison between the extracts was not provided. The superior effect showed by DHA may be due to its smaller three-dimensional molecular structure, which may facilitate its diffusion through the biofilm extracellular matrix.

The mechanisms that underline the observed bacterial inhibitory effect of EPA or DHA against oral pathogens are still unknown. Possibly—as described for other PUFAs—the fatty acids are incorporated into the cell plasma membrane, resulting in greater membrane fluidity and permeability, thus affecting its integrity, which leads to cell death [23,29]. This effect could be exacerbated by the presence of unsaturated double bonds that exert a direct toxic effect on the bacterial cell membrane [25]. At concentrations of 100 μM, DHA and EPA are not cytotoxic to human oral tissue cells [37], or to C2C12 myoblasts [58], but maintain their antibacterial activity. In the present investigation, we have used a concentration of 100 μM and the SEM observations have shown not only a reduction in the number of bacteria present, but also distinct morphological changes with observable structural damage. These observations coincide with previous observations, also using 100 μM of EPA and DHA [37]. In addition, similar events have been described for other bacterial species upon exposure to DHA, such as *Burkholderia cenocepacia*, [28], or *Helicobacter pylori* [25].

The quantitative analysis showed that EPA and DHA, when compared to the controls, significantly reduced the bacterial viability of the tested species (*p* < 0.001). In the case of *V. parvula*, *F. nucleatum* or *P. gingivalis*, they showed greater antimicrobial activity than CHX, an antiseptic considered as the gold standard for use in oral mouthwash formulations [18]. The effect of the tested extracts in reducing cell viability was also observed in the CLSM analysis. The discs treated with both DHA and EPA, although having a more pronounced effect with DHA, showed a low cell viability. These CLSM results were fully congruent with the qPCR results. It was noticeable the higher antimicrobial effect of DHA and EPA, when compared with CHX, has shown a marked antibacterial effect in multispecies biofilm models, both in vitro and in vivo [40,59]. One of the reported advantages of CHX molecules is their binding capability to oral tissues, which allows its slow and continuous release [60,61] for up to 12 h (substantivity) [62]. The possible substantivity and pharmacokinetics of EPA and DHA extracts are currently unknown, and therefore further investigations are required before considering these extracts as real alternatives to currently used oral antiseptics.

Although the study is the first to demonstrate the antibacterial effect of EPA or DHA extracts in a validated multi-species in vitro biofilm model [38], there are some limitations that should be highlighted. Firstly, only six bacterial species were used, in comparison with the hundreds present in naturally occurring subgingival biofilms. Furthermore, the model used represents a static biofilm model, and in its current design, the effect of the immune system cells is not involved. Although this model has a mixed population of six bacterial strains representing the initial, early and late colonizers present in subgingival biofilms, it does not have the diversity usually found in natural subgingival biofilms, as these can reach 200 species. Our model also only includes one strain of each bacterial species, which can provide different results, especially when regarding eventual evolutionary mechanisms of response to the PUFAs. In regard to the extracts analysed, their commercial formulation entails their dissolution in ethanol at different concentrations. In order to rule out the possibility that the potential antiseptic effect of these agents was due to ethanol, rather than to the extracts tested, ethanol was included in the experiments as a control (at the same concentration found in each of the commercial products), thus demonstrating the specific antibacterial effect of the PUFA extracts. On the other hand, the findings of this research are expected to be comparable to other artificial surfaces used in dentistry, for example titanium and zirconia [40].

Besides the presented limitations of the study, this kind of models should be regarded as a primary step in the research process, identifying possible candidate molecules to be investigated in studies with a higher level of scientific evidence.

## 5. Conclusions

The results of the present study provide evidence that EPA or DHA extracts demonstrated a relevant antibacterial capacity against six bacterial species in a validated in vitro biofilm model. Specifically, EPA or DHA extracts (at 100 μM) resulted in statistically significant reductions in the CFU mL^−1^ of *S. oralis*, *A. naeslundii*, *V. parvula*, *F. nucleatum*, *P. gingivalis* and *A. actinomycetemcomitans* after 60 s of exposure.

Further research is needed in order to evaluate the possible use of PUFAs in the chemical plaque control during the management of periodontitis.

## Figures and Tables

**Figure 1 nutrients-12-02812-f001:**
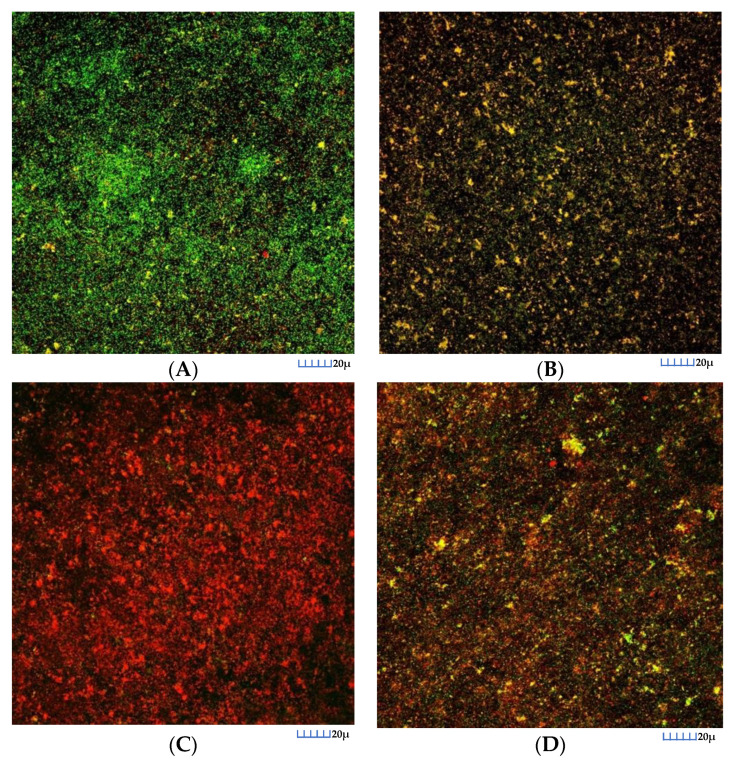
Maximum projection of images obtained by confocal laser scanning microscopy (CLSM) of the 72 h biofilms, where the growth of these biofilms was observed on the surfaces of the hydroxyapatite discs after 60 s of exposure: (**A**) to the negative control (phosphate buffer saline); (**B**) to the ethanol solution; (**C**) to the docosahexaenoic acid (DHA) extracts (100 μM concentration) and (**D**) to 0.2% chlorhexidine. Specimens were stained with the LIVE/DEAD^®^ BacLightTM Bacterial Viability Kit solution, containing SYTO 9 and Propidium Iodide nucleic acid stains. Cells with a compromised membrane that are considered to be dead or dying were stain red (PI), whereas cells with an intact membrane were stain green (SYTO9).

**Figure 2 nutrients-12-02812-f002:**
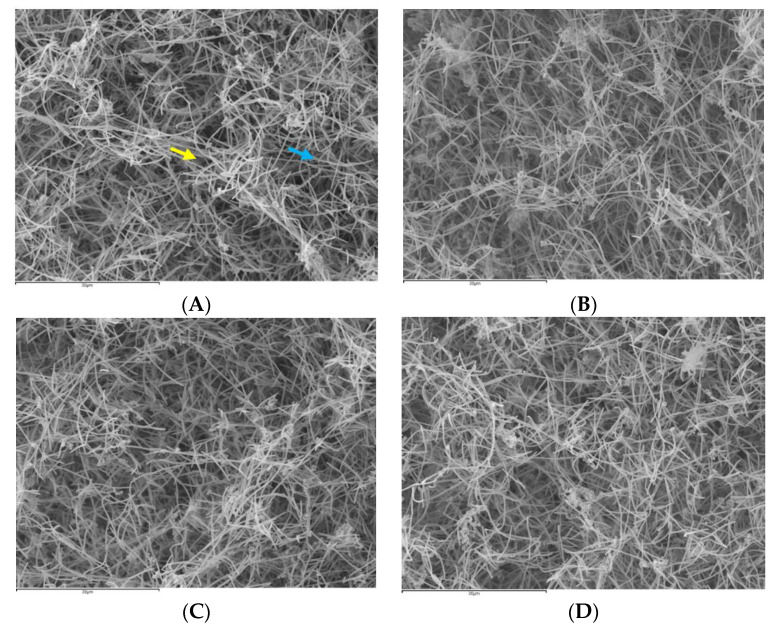
Scanning electron microscopy (SEM) of biofilms with an evolution of 72 h in hydroxyapatite (HA) discs treated with the negative control: phosphate buffer saline (PBS) (**A**), with docosahexaenoic acid (DHA) at 100 µM (**B**), with EtOH (**C**) or with the positive control: 0.2% chlorhexidine (CHX)(**D**). A dense bacterial population could be observed on the HA discs treated with PBS (**A**), forming discontinuous layers of bacteria bonded to the discs. Meanwhile, on the biofilms of the discs treated with DHA (**B**), a lower density of cells distributed across the surface of the disc could be seen, and some of these exhibited structural damages. Likewise, on the discs treated with EtOH (**C**) or CHX (**D**), a reduction in the bacterial density present on the surface of the disc could also be observe, although it was lower than that on the discs treated with DHA (**B**). Chains of *Aggregatibacter* and/or *Streptococcus* (blue arrow) and fusiform bacilli of the *F. nucleatum* genus (yellow arrow) could be identified. Magnification (**A**–**D**): 1500×. The samples were dried by critical points and coated with gold by sputtering.

**Figure 3 nutrients-12-02812-f003:**
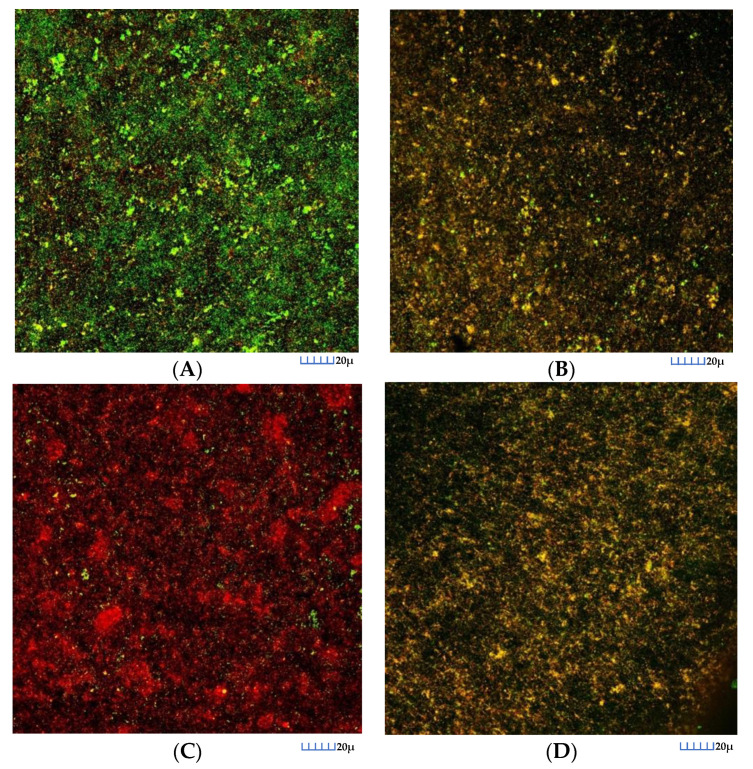
Maximum projection of images obtained by confocal laser scanning microscopy (CLSM) of the 72 h biofilms, where the growth of these biofilms was observed on the surfaces of the hydroxyapatite discs, stained with LIVE/DEAD^®^ BacLightTM Bacterial Viability Kit, after 60 s of exposure: (**A**) to the negative control (phosphate buffer saline); (**B**) to the ethanol solution; (**C**) to the docosahexaenoic acid (EPA) extracts (100 μM concentration) and (**D**) to 0.2% chlorhexidine. Specimens were stained with the LIVE/DEAD^®^ BacLightTM Bacterial Viability Kit solution, containing SYTO 9 and Propidium Iodide nucleic acid stains. Cells with a compromised membrane that are considered to be dead or dying were stained red (PI), whereas cells with an intact membrane were stained green (SYTO9).

**Figure 4 nutrients-12-02812-f004:**
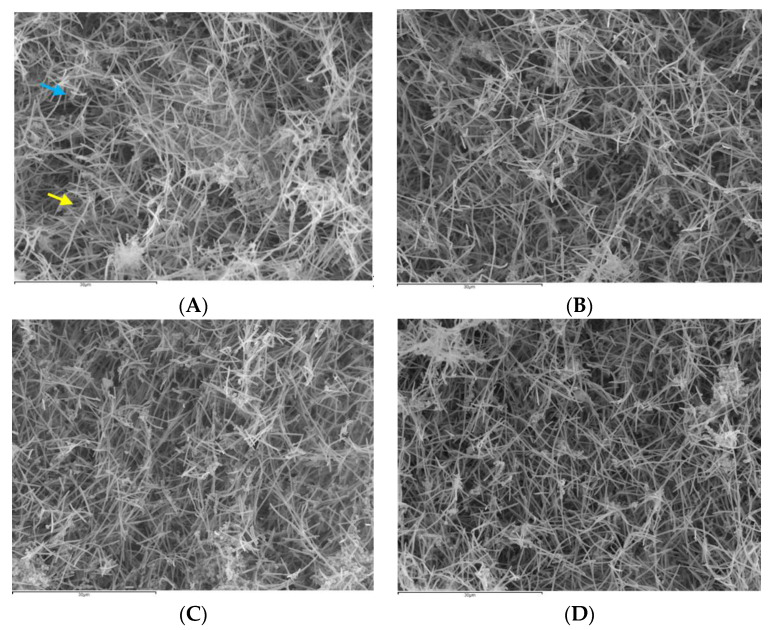
Scanning electron microscopy (SEM) of biofilms with an evolution of 72 h in hydroxyapatite (HA) discs treated with the negative control: phosphate buffer saline (PBS) (**A**), with docosahexaenoic acid (EPA) at 100 µM (**B**), with ethanol (**C**) or with the positive control: 0.2% chlorhexidine (CHX)(**D**). A dense bacterial population could be observed on the HA discs treated with PBS (**A**), forming discontinuous layers of bacteria bonded to the discs. Meanwhile, on the biofilms of the discs treated with EPA (**B**), a lower density of cells distributed across the surface of the disc could be seen, and some of these exhibited structural damages. Likewise, on the discs treated with EtOH (**C**) or CHX (**D**), a reduction could also be observed in the bacterial density present on the surface of the disc, although this reduction was slighter than that on the discs treated with EPA (**B**). Chains of *Aggregatibacter* and/or *Streptococcus* (blue arrow) and fusiform bacilli of the *F. nucleatum* genus (yellow arrow) could be identified. Magnification (**A**–**D**): 1500×. The samples were dried by critical points and coated with gold by sputtering.

**Table 1 nutrients-12-02812-t001:** Antibacterial effects of docosahexaenoic acid (DHA), as observed in the mean number of viable bacteria counts [colony forming units per mL (CFUs mL^−1^), determined by quantitative polymerase chain reaction], evaluated in an in vitro multi-species biofilm model. Data are expressed as mean and standard deviation (SD). Differences are considered statistically significant at *p* ≤ 0.05. PBS: phosphate buffer saline; DHA: 100 μM docosahexaenoic acid; EtOH: ethanol at the same concentration as the present in DHA; CHX: 0.2% chlorhexidine.

	Treatments	Mean (SD)	Global *p*	% of Reduction of Viable CFUs mL^−1^ as Compared with PBS
*S. oralis*	PBS	7.80 × 10^7^ (3.46 × 10^7^)	<0.001	
	DHA	3.49 × 10^4^ (4.53 × 10^3^)		99.96%
	EtOH	1.99 × 10^7^ (1.23 × 10^7^)		74.49%
	CHX	1.39 × 10^6^ (3.81 × 10^5^)		98.22%
*A. naeslundii*	PBS	3.22 × 10^6^ (1.04 × 10^6^)	<0.001	
	DHA	5.88 × 10^3^ (1.51 × 10^3^)		99.82%
	EtOH	1.58 × 10^6^ (9.42 × 10^5^)		51.04%
	CHX	3.23 × 10^5^ (1.88 × 10^5^)		90.07%
*V. parvula*	PBS	4.56 × 10^7^ (9.86 × 10^6^)	<0.001	
	DHA	8.83 × 10^4^ (2.60 × 10^4^)		99.80%
	EtOH	1.68 × 10^7^ (5.40 × 10^6^)		63.16%
	CHX	8.18 × 10^6^ (9.59 × 10^6^)		82.07%
*F. nucleatum*	PBS	2.24 × 10^6^ (1.04 × 10^6^)	<0.001	
	DHA	2.01 × 10^3^ (8.68 × 10^2^)		99.91%
	EtOH	7.36 × 10^5^ (6.70 × 10^5^)		67.15%
	CHX	1.62 × 10^6^ (1.15 × 10^6^)		27.68%
*P. gingivalis*	PBS	2.32 × 10^7^ (8.31 × 10^6^)	<0.001	
	DHA	1.77 × 10^4^ (2.71 × 10^3^)		99.92%
	EtOH	2.37 × 10^6^ (1.20 × 10^6^)		89.79%
	CHX	1.82 × 10^6^ (3.13 × 10^6^)		92.16%
*A. actinomycetemcomitans*	PBS	1.14 × 10^7^ (4.84 × 10^6^)	<0.001	
	DHA	1.12 × 10^4^ (4.71 × 10^3^)		99.90%
	EtOH	4.62 × 10^6^ (2.25 × 10^6^)		59.48%
	CHX	2.10 × 10^6^ (5.64 × 10^5^)		81.58%

**Table 2 nutrients-12-02812-t002:** Comparisons between docosahexaenoic acid (DHA) and the controls used as observed in the mean number of viable bacteria counts (colony forming units per mL (CFUs mL^−1^), determined by quantitative polymerase chain reaction) evaluated in an in vitro multi-species biofilm model. Differences are considered statistically significant at *p* ≤ 0.05. PBS: phosphate buffer saline; DHA: 100 μM docosahexaenoic acid; EtOH: ethanol at the same concentration as the present in DHA; CHX: 0.2% chlorhexidine.

	Comparisons	Mean Difference	95% Confidence Interval for Difference		Post Hoc *p*
			Lower Bound	Upper Bound	
*S. oralis*	PBS-DHA	7.80 × 10^7^	5.36 × 10^7^	1.02 × 10^8^	<0.001
	PBS-EtOH	5.81 × 10^7^	3.37 × 10^7^	8.25 × 10^7^	<0.001
	PBS-CHX	7.66 × 10^7^	5.23 × 10^7^	1.01 × 10^8^	<0.001
	EtOH-DHA	1.99 × 10^7^	−4.47 × 10^6^	4.43 × 10^7^	0.170
	CHX-DHA	1.35 × 10^6^	−2.30 × 10^7^	2.57 × 10^7^	1.000
	EtOH-CHX	1.86 × 10^7^	−5.82 × 10^6^	4.29 × 10^7^	0.240
*A. naeslundii*	PBS-DHA	3.22 × 10^6^	2.28 × 10^6^	4.15 × 10^6^	<0.001
	PBS-EtOH	1.64 × 10^6^	7.05 × 10^5^	2.58 × 10^6^	<0.001
	PBS-CHX	2.90 × 10^6^	1.96 × 10^6^	3.84 × 10^6^	<0.001
	EtOH-DHA	1.57 × 10^6^	6.31 × 10^5^	2.51 × 10^6^	<0.001
	CHX-DHA	3.17 × 10^5^	−6.23 × 10^5^	1.26 × 10^6^	1.000
	EtOH-CHX	1.25 × 10^6^	3.14 × 10^5^	2.19 × 10^6^	0.004
*V. parvula*	PBS-DHA	4.55 × 10^7^	3.57 × 10^7^	5.53 × 10^7^	<0.001
	PBS-EtOH	2.88 × 10^7^	1.90 × 10^7^	3.86 × 10^7^	<0.001
	PBS-CHX	3.74 × 10^7^	2.76 × 10^7^	4.72 × 10^7^	<0.001
	EtOH-DHA	1.67 × 10^7^	6.89 × 10^6^	2.65 × 10^7^	<0.001
	CHX-DHA	8.09 × 10^6^	−1.70 × 10^6^	1.79 × 10^7^	0.160
	EtOH-CHX	8.59 × 10^6^	−1.21 × 10^6^	1.84 × 10^7^	0.115
*F. nucleatum*	PBS-DHA	2.24 × 10^6^	1.12 × 10^6^	3.36 × 10^6^	<0.001
	PBS-EtOH	1.50 × 10^6^	3.86 × 10^5^	2.62 × 10^6^	0.004
	PBS-CHX	6.15 × 10^5^	−5.03 × 10^5^	1.73 × 10^6^	0.789
	EtOH-DHA	7.34 × 10^5^	−3.84 × 10^5^	1.85 × 10^6^	0.445
	CHX-DHA	1.62 × 10^6^	5.05 × 10^5^	2.74 × 10^6^	0.002
	EtOH-CHX	−8.89 × 10^5^	−3.84 × 10^5^	1.85 × 10^6^	0.194
*P. gingivalis*	PBS-DHA	2.32 × 10^7^	1.73 × 10^7^	2.92 × 10^7^	<0.001
	PBS-EtOH	2.09 × 10^7^	1.49 × 10^7^	2.68 × 10^7^	<0.001
	PBS-CHX	2.14 × 10^7^	1.55 × 10^7^	2.73 × 10^7^	<0.001
	EtOH-DHA	2.36 × 10^6^	−3.58 × 10^6^	8.29 × 10^6^	1.000
	CHX-DHA	1.81 × 10^6^	−4.13 × 10^6^	7.74 × 10^6^	1.000
	EtOH-CHX	5.50 × 10^5^	−5.39 × 10^6^	6.49 × 10^6^	1.000
*A. actinomycetemcomitans*	PBS-DHA	1.14 × 10^7^	7.81 × 10^6^	1.49 × 10^7^	<0.001
	PBS-EtOH	6.76 × 10^6^	3.20 × 10^6^	1.03 × 10^7^	<0.001
	PBS-CHX	9.28 × 10^6^	5.73 × 10^6^	1.28 × 10^7^	<0.001
	EtOH-DHA	4.61 × 10^6^	1.06 × 10^6^	8.17 × 10^6^	0.006
	CHX-DHA	2.08 × 10^6^	−1.47 × 10^6^	5.64 × 10^6^	0.654
	EtOH-CHX	2.53 × 10^6^	−1.03 × 10^6^	6.08 × 10^6^	0.325

**Table 3 nutrients-12-02812-t003:** Antibacterial effects of eicosapentaenoic acid (EPA) as observed in the mean number of viable bacteria counts (colony-forming units per mL (CFUs mL^−1^), determined by quantitative polymerase chain reaction), evaluated in an in vitro multi-species biofilm model. Data are expressed as mean and standard deviation (SD). Differences are considered significant at *p* ≤ 0.05. PBS: phosphate buffer saline; EPA: 100 μM eicosapentaenoic acid; EtOH: ethanol at the same concentration as that present in EPA; CHX: 0.2% chlorhexidine.

	Treatments	Mean (SD)	Global *p*	% of Reduction in Viable CFUs mL^−1^ as Compared with PBS	
*S. oralis*	PBS	4.71 × 10^7^ (1.38 × 10^7^)	<0.001	
	EPA	1.34 × 10^6^ (5.12 × 10^5^)		97.16%
	EtOH	9.17 × 10^6^ (2.46 × 10^6^)		80.54%
	CHX	1.64 × 10^6^ (7.58 × 10^5^)		96.52%
*A. naeslundii*	PBS	3.63 × 10^6^ (1.47 × 10^6^)	<0.001	
	EPA	5.98 × 10^4^ (1.82 × 10^4^)		98.36%
	EtOH	6.08 × 10^5^ (1.91 × 10^5^)		83.25%
	CHX	3.66 × 10^5^ (1.30 × 10^5^)		89.92%
*V. parvula*	PBS	6.43 × 10^7^ (1.66 × 10^7^)	<0.001	
	EPA	2.94 × 10^6^ (8.30 × 10^5^)		95.43%
	EtOH	2.10 × 10^7^ (3.14 × 10^6^)		67.34%
	CHX	1.80 × 10^7^ (1.08 × 10^7^)		72.01%
*F. nucleatum*	PBS	2.16 × 10^6^ (9.67 × 10^5^)	<0.001	
	EPA	2.88 × 10^4^ (1.45 × 10^4^)		98.67%
	EtOH	6.66 × 10^5^ (5.88 × 10^5^)		69.17%
	CHX	1.24 × 10^6^ (4.58 × 10^5^)		42.60%
*P. gingivalis*	PBS	1.27 × 10^7^ (1.60 × 10^6^)	<0.001	
	EPA	3.16 × 10^5^ (1.55 × 10^5^)		97.51%
	EtOH	1.92 × 10^6^ (5.04 × 10^5^)		84.88%
	CHX	1.63 × 10^6^ (2.18 × 10^5^)		87.17%
*A. actinomycetemcomitans*	PBS	6.84 × 10^6^ (3.40 × 10^6^)	<0.001	
	EPA	5.91 × 10^5^ (3.18 × 10^5^)		91.36%
	EtOH	2.07 × 10^6^ (7.91 × 10^5^)		69.74%
	CHX	2.29 × 10^6^ (1.62 × 10^6^)		66.52%

**Table 4 nutrients-12-02812-t004:** Comparisons between eicosapentaenoic acid (EPA) and the controls used as observed in the mean number of viable bacteria counts (colony-forming units per mL (CFUs mL^−1^), determined by quantitative polymerase chain reaction), evaluated in an in vitro multi-species biofilm model. Differences were considered significant at *p* ≤ 0.05. PBS: phosphate buffer saline; EPA: 100 μM eicosapentaenoic acid; EtOH: ethanol at the same concentration as the present in EPA; CHX: 0.2% chlorhexidine.

	Comparisons	Mean Difference	95% Confidence Interval for Difference		Post Hoc *p*
			Lower Bound	Upper Bound	
*S. oralis*	PBS-EPA	4.57 × 10^7^	3.64 × 10^7^	5.50 × 10^7^	<0.001
	PBS-EtOH	3.79 × 10^7^	2.86 × 10^7^	4.72 × 10^7^	<0.001
	PBS-CHX	4.54 × 10^7^	3.61 × 10^7^	5.47 × 10^7^	<0.001
	EtOH-EPA	7.83 × 10^6^	−1.47 × 10^6^	1.71 × 10^7^	0.144
	CHX-EPA	2.96 × 10^5^	−9.00 × 10^6^	9.59 × 10^6^	1.000
	EtOH-CHX	7.53 × 10^6^	−1.76 × 10^6^	1.68 × 10^7^	0.177
*A. naeslundii*	PBS-EPA	3.57 × 10^6^	2.59 × 10^6^	4.56 × 10^6^	<0.001
	PBS-EtOH	3.02 × 10^6^	2.04 × 10^6^	4.01 × 10^6^	<0.001
	PBS-CHX	3.27 × 10^6^	2.28 × 10^6^	4.25 × 10^6^	<0.001
	EtOH-EPA	5.48 × 10^5^	−4.39 × 10^5^	1.54 × 10^6^	0.768
	CHX-EPA	3.06 × 10^5^	−6.81 × 10^5^	1.29 × 10^6^	1.000
	EtOH-CHX	2.42 × 10^5^	−7.45 × 10^5^	1.23 × 10^6^	1.000
*V. parvula*	PBS-EPA	6.14 × 10^7^	4.81 × 10^7^	7.47 × 10^7^	<0.001
	PBS-EtOH	4.33 × 10^7^	3.00 × 10^7^	5.66 × 10^7^	<0.001
	PBS-CHX	4.63 × 10^7^	3.30 × 10^7^	5.96 × 10^7^	<0.001
	EtOH-EPA	1.81 × 10^7^	4.82 × 10^6^	3.14 × 10^7^	0.003
	CHX-EPA	1.51 × 10^7^	1.81 × 10^6^	2.84 × 10^7^	0.019
	EtOH-CHX	3.00 × 10^6^	−1.03 × 10^7^	1.63 × 10^7^	1.000
*F. nucleatum*	PBS-EPA	2.13 × 10^6^	1.32 × 10^6^	2.94 × 10^6^	<0.001
	PBS-EtOH	1.50 × 10^6^	6.88 × 10^5^	2.31 × 10^6^	<0.001
	PBS-CHX	9.27 × 10^5^	1.18 × 10^5^	1.74 × 10^6^	0.018
	EtOH-EPA	6.37 × 10^5^	−1.72 × 10^5^	1.45 × 10^6^	0.205
	CHX-EPA	1.21 × 10^6^	3.98 × 10^5^	2.02 × 10^6^	0.001
	EtOH-CHX	−5.70 × 10^5^	−1.38 × 10^6^	2.39 × 10^5^	0.336
*P. gingivalis*	PBS-EPA	1.23 × 10^7^	1.12 × 10^7^	1.35 × 10^7^	<0.001
	PBS-EtOH	1.07 × 10^7^	9.61 × 10^6^	1.19 × 10^7^	<0.001
	PBS-CHX	1.10 × 10^7^	9.90 × 10^6^	1.21 × 10^7^	<0.001
	EtOH-EPA	1.60 × 10^6^	4.77 × 10^5^	2.73 × 10^6^	0.002
	CHX-EPA	1.32 × 10^6^	1.93 × 10^5^	2.44 × 10^6^	0.014
	EtOH-CHX	2.84 × 10^5^	−8.41 × 10^5^	1.41 × 10^6^	1.000
*A. actinomycetemcomitans*	PBS-EPA	6.25 × 10^6^	3.70 × 10^6^	8.81 × 10^6^	<0.001
	PBS-EtOH	4.77 × 10^6^	2.21 × 10^6^	7.33 × 10^6^	<0.001
	PBS-CHX	4.55 × 10^6^	2.00 × 10^6^	7.11 × 10^6^	<0.001
	EtOH-EPA	1.48 × 10^6^	−1.07 × 10^6^	4.04 × 10^6^	0.676
	CHX-EPA	1.70 × 10^6^	−8.55 × 10^5^	4.26 × 10^6^	0.423
	EtOH-CHX	−2.18 × 10^5^	−2.77 × 10^6^	2.34 × 10^6^	1.000

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
