# Peer review of "Antimicrobial Activity of EPA and DHA against Oral Pathogenic Bacteria Using an In Vitro Multi-Species Subgingival Biofilm Model"

_nutrients, 2020, doi:10.3390/nu12092812_

Round 1
Reviewer 1 Report
Comments to the authors:
This manuscript is aimed at the investigation to evaluate the antimicrobial activity of the omega-3 FAs DHA and EPA in six oral pathogenic bacteria. Ribeiro-Vidal et al. determined that both DHA and EPA were bactericidal by employing quantitative polymerase chain reaction (qPCR), confocal laser scanning microscopy (CLSM), and scanning electron microscopy (SEM). I believe that this manuscript is relevant to the audience that reads this journal and provides essential and useful information regarding fatty acid research. However, I have some concerns that need to be addressed.
Main concerns:
- In the material and methods section (Pag. 2, Lines 74-78), the authors do not specify whether PUFAs extracts from fish oil only contain either EPA or DHA or if these extracts also contain other FAs in the ethanolic solution. Pure EPA or DHA can be purchased from Sigma Aldrich; why is it necessary to use an ethanolic solution of these PUFAs? What is the % of ethanol (or amount of ethanol) that is required for solubilizing the above-mentioned PUFAs without affecting the antibacterial activity of the six bacterial strains? Why not DMSO? This information is vital for reproducibility reasons.
- How do the authors determine that 100 mM of either DHA or EPA is the appropriated concentration for their antimicrobial activity? (Page 3, Lines 97-99). Is this value the MIC value obtained from previous experiments involving the biofilm approach? This aspect is not clear.
- Performing only two qPCR experiments is not acceptable. For experiments involving pathogenic bacteria require at least three independent biological replicates (Page 3, Line 120).
- It is required that authors provide information regarding the biological strains that were used in this study in separated subsection (i. e. ATCC Number, how these bacterial were growth and the media that were used for culturing these bacteria).
- On page 4, Line 145, the author did not explain what type of data was averaged. This reviewer infers that the data that will be averaged is the number of bacterial cells that were stained in green and red. If this reviewer is correct in his interpretation, how these cells were counted, avoiding human error? Was the green/red ratio counted by using the Fiji software? Why did the author not use flow cytometry for these experiments?
- This reviewer understands that biofilm data need to be compared with microdilution susceptibility tests and explain potential differences in the results.
Minors concerns:
- Authors need to use “italic letters” for naming bacteria.
Author Response
Main Concerns:
1. In the material and methods section (Pag. 2, Lines 74-78), the authors do not specify whether PUFAs extracts from fish oil only contain either EPA or DHA or if these extracts also contain other FAs in the ethanolic solution. Pure EPA or DHA can be purchased from Sigma Aldrich; why is it necessary to use an ethanolic solution of these PUFAs? What is the % of ethanol (or amount of ethanol) that is required for solubilizing the above-mentioned PUFAs without affecting the antibacterial activity of the six bacterial strains? Why not DMSO? This information is vital for reproducibility reasons.
The PUFAS used in the present study are pure extracts acquired to Sigma Aldrich that contain only EPA or DHA. Following the manufacturer recommendations, we acquired these extracts solubilized in ethanol with solutions of EPA and DHA at concentrations of 500.0 ± 2.8 μg/mL and 499.9 ± 2.7 μg/mL, respectively. Again, following the manufacturer instructions, we diluted the alcoholic solution in water to obtain the different concentrations of the PUFAs used in the antimicrobial assays.
The same concentration of ethanol indicated by the manufacturer for the solubilization was used as control in all the assays in order to assess whether the antimicrobial effect was due to the ethanol or to the respective fatty acid.
Following the reviewer suggestion, the section of material and methods was improved in the revised version of the manuscript. Where in the page 2 lines 74-75 was read …. “The unsaturated PUFAs extracts from fish oil independently evaluated in this investigation were EPA and DHA, solubilised in ethanol (Cerilliant®, Sigma-Aldrich, Barcelona, Spain),….” it now reads in the page 2 lines 80-81:
“The PUFAs independently evaluated in this investigation were EPA and DHA, obtained already solubilized in ethanol (Cerilliant®, Sigma-Aldrich, Barcelona, Spain)”
2. How do the authors determine that 100 mM of either DHA or EPA is the appropriated concentration for their antimicrobial activity? (Page 3, Lines 97-99). Is this value the MIC value obtained from previous experiments involving the biofilm approach? This aspect is not clear.
In fact, before the start of experiments using the subgingival biofilm model, we assessed whether the tested PUFAs had any antimicrobial effect on the bacterial strains included in the model, testing each fatty acid independently with each bacterial strain in planktonic state. These data were not included in the manuscript to avoid overloading the paper with preliminar data, but in light of the query from the reviewer we have included in the revised version new sections in material and methods and results. Once we had the results from the planktonic tests, 100μM concentration was selected for the biofilm assays.
The following text in the 2.1 section, page 2 lines 75-78 was deleted “both at a concentration of 100 μM, based on the Minimum Bactericidal Concentrations (MBC) and the Minimum Inhibitory Concentrations (MIC) of EPA and DHA for the six bacterial species used in the biofilm model.”.
A new 2.3 section, page 3 lines 112-134, was created in the manuscript with the following title “Antibacterial effect of EPA and DHA against planktonic bacteria”. In this new section, the following text can be read:
“ For determining which concentration of each of EPA and DHA were appropriate for the biofilm model assays, we undertook independent previous microtiter plate-based antibacterial assays for each of the studied fatty acids. In brief, pure cultures of the 6 selected bacterial strains were grown anaerobically in a protein rich medium containing brain‐heart infusion (BHI) (Becton, Dickinson and Company, Franklin Lakes, NJ, USA) supplemented with 2.5 g L−1 mucin (Oxoid), 1.0 g L−1 yeast extract (Oxoid), 0.1 g L−1 cysteine (Sigma), 2.0 g L−1 sodium bicarbonate (Merck), 5.0 mg L−1 hemin (Sigma), 1.0 mg L−1 menadione (Merck), and 0.25% (v/v) glutamic acid (Sigma). At mid‐exponential phase of bacterial growth (measured by spectrophotometry), bacteria were placed on a 96‐wells microtitre plates adding 200μL of a mixture of each bacteria inoculum at a final concentration of 106 colony forming units (CFUs) mL−1, and EPA or DHA for a final concentration of 12.5, 25, 50, 100 and 200μM. Plates had a set of controls: Phosphate buffered saline (PBS) was used as negative control; ethanol controls (adjusted to match the ethanol concentration present in each of the fatty acids (EtOH)); positive control (bacteria without any treatment). A measurement (optical density, O.D.595) as t = 0 absorbance was taken in a microtitre plate reader (Optic Ivymen System 2100‐C; I.C.T.; La Rioja, Spain). The microplates were incubated for 48 h at 37 °C under anaerobic conditions, and absorbance was measured each 2 h, in order to determine the bacterial growth until reaching a stationary growth phase. MIC (minimum inhibitory concentration) and MBC (minimum bactericidal concentration) values were calculated and confirmed by microbial plate counting on blood agar media. Accordingly, the lowest concentration of the DHA or EPA showing growth inhibition was considered as the MIC, whereas the lowest concentration of the DHA or EPA that showed zero growth in blood agar plates, after spot inoculation and incubation for 72 h, was recorded as the MBC. All experiments were performed in triplicate with appropriate controls.”
A new results 3.1 section, page 5 lines 239-247, was created in the manuscript with the following title “Antibacterial effect of EPA and DHA against planktonic bacteria”. In this new section, the following text can be read:
“MICs and MBCs values against the six bacterial strains selected in planktonic state were determined for each of the fatty acids. In the case of DHA, the MICs were 50 μM for S. oralis, A. naeslundii and V. parvula; 100 μM for F. nucleatum; and 25 μM for P. gingivalis and A. actinomycetemcomitans. The MBCs were 100 μM for all the six bacterial species.
In the case of EPA, the MICs were 50 μM for S. oralis and V. parvula and; 100 μM for A. naeslundii and F. nucleatum; and 25 μM for P. gingivalis and A. actinomycetemcomitans. The MBCs were 100 μM for all the six bacterial species. Based on these results, the 100 μM concentration was selected in the biofilm experiments with the two fatty acids.”
3. Performing only two qPCR experiments is not acceptable. For experiments involving pathogenic bacteria require at least three independent biological replicates (Page 3, Line 120).
Effectively, the experiments were repeated three times, as it is specified in the page 3, lines 103-104: …” All the independent sets of experiments for each of the PUFAs were repeated three times on different days using fresh bacterial cultures with trios of biofilms for each analysis”….
In the page 3, line 120, what is indicated is the number of times that each sample was put on the PCR plate in order to ensure the result and, only in the case of discrepancy between the two results, the procedure was repeated.
4. It is required that authors provide information regarding the biological strains that were used in this study in separated subsection (i. e. ATCC Number, how these bacterial were growth and the media that were used for culturing these bacteria).
With the objective of avoiding the overload of the manuscript, these data was included in the following way “A multispecies in vitro biofilm model was developed, as previously described by Sánchez and co-workers [35]” (Pg2. Lines 80-81). But following the reviewer recommendations, a new 2.2 section, page 2 lines 82-88, in material and methods was included with the following title “. Bacterial strains and culture conditions”. In this new section, the following text can be read:
“Reference strains of Streptococcus oralis CECT 907T, Veillonella parvula NCTC 11810, Actinomyces naeslundii ATCC 19039, Fusobacterium nucleatum DMSZ 20482, Aggregatibacter actinomycetemcomitans DSMZ 8324, and P. gingivalis ATCC 33277 were used. These bacteria were grown on blood agar plates (Blood Agar Oxoid No 2; Oxoid, Basingstoke, UK), supplemented with 5% (v/v) sterile horse blood (Oxoid), 5.0 mg L−1 hemin (Sigma, St. Louis, MO, USA) and 1.0 mg L−1 menadione (Merck, Darmstadt, Germany) in anaerobic conditions (10% H2, 10% CO2, and balance N2) at 37 °C for 24–72 h.”
5. On page 4, Line 145, the author did not explain what type of data was averaged. This reviewer infers that the data that will be averaged is the number of bacterial cells that were stained in green and red. If this reviewer is correct in his interpretation, how these cells were counted, avoiding human error? Was the green/red ratio counted by using the Fiji software? Why did the author not use flow cytometry for these experiments?
Effectively, the Fiji software was used for calculating the area occupied by live and death cells in the biofilm. We did not use the flow cytometry technique since the biofilm it is a mixed population of bacterial strains, with substantially different morphology, as in the specific case of Fusobacterium nucleatum, and therefore, flow cytometry would not be appropriate.
6. This reviewer understands that biofilm data need to be compared with microdilution susceptibility tests and explain potential differences in the results.
Following the reviewer's suggestion, the CMI and CMB methodology as well as the test results are presented in the new 2.3 material and methods section (page 3, lines 112-134) and in the new 3.1 results section (page 5, lines 242-250).
Minor Concerns:
1. Authors need to use “italic letters” for naming bacteria.
Following the reviewer's indication, the full text has been revised to correct this error.
Reviewer 2 Report
The authors have presented a novel approach to utilize DHA and EPA as direct antimicrobial agents to to oral microbial species. To improve the quality, minor reviews suggested.
Introduction:
-Periodontal dysbiosis is not caused by Porphyromonas gingivalis. The authors need to clarify the definition of dysbiosis as the molecular drive prior to tissue damage. In fact, an ecological switch happens to the oral microbiome prior to tissue phenotypes. Please include metabolic concepts of the dysbiosis as well as community approach versus monogenic assumptions. Provide citations of the complexity of both planktonic and biofilm ecological signatures and introduce supra and subgingival species.
-Isolated molecules are not being studied here, but rather crude extract from fish oil to form EPA and DHA, not specialized mediators such as lipoxins, resolvins etc. Please clarify (line 70-71; page 2).
If EPA and DHA have inhibitory functions to both gram-positive and gram-negative, why are the authors continuing to study? IS that due to lack of MOI, concentration, type of bacteria? Please clarify?
Methods:
- Time point issues. What is the justification of 70h application on the biofilms? Was there a time point or concentration dependent pilot study done before this decision? Please justify.
- Concentration versus toxicity: Please provide a justification or reference. High concentration of most molecules will cause toxic effects to cells and bacterial cells. We need to understand how the authors selected the concentration?
Results:
-Excessive tables are confusing and comparisons are not appropriate. Please provide a histogram or heatmap comparing DHA vs EPA and controls side-by-side.
- Confocal quantification missing. Confocal microscopy provide layers of X-stack images that are then compiled to a 2D figure presented on figure 1 and 3. the authors need to add MFI values of these images. MFI is mean or median fluorescence intensity to allow for evaluations on the quality and accuracy of the imaging. Supplementary or main data is fine.
- Lack of labels in the figures. Both SEM and CLSM would benefit from having appropriate stainning labels. For example for green staining you add add a black box inside the photo and write in green the word FITC. this can help the reader.
- Lack of scale bars to the imaging figures. Please add scale bars to all imaging figures.
Discussion:
-Please discuss the behavior of biofilms on multiple materials, including crowns, titanium implants.
-It is important to acknowledge that this is a monogenic biofilm and the artificial study is limited. Naturally polygenic, the oral biofilm would have evolutionary mechanisms of response that need to be discussed in the context of your data.
- Conclusions are well written, but need to provide a sentence linking to periodontal disease as was portraid in the introduction.
Author Response
Introduction:
1. Periodontal dysbiosis is not caused by Porphyromonas gingivalis. The authors need to clarify the definition of dysbiosis as the molecular drive prior to tissue damage. In fact, an ecological switch happens to the oral microbiome prior to tissue phenotypes. Please include metabolic concepts of the dysbiosis as well as community approach versus monogenic assumptions. Provide citations of the complexity of both planktonic and biofilm ecological signatures and introduce supra and subgingival species.
The reviewer is right as the text present in the manuscript might mislead the reader to understand that P. gingivalis directly leads to dysbiosis. In fact, although it is considered a keystone pathogen for periodontitis it is just a small piece of all the dysbiosis process that characterize periodontitis.
Basing our understating of periodontitis in the polymicrobial synergy and dysbiosis model proposed by Hajishengallis and coworkers in 2012, what we aimed to explain to the readers was that contrarily to the past understanding on the role of specific “periopathogens” as direct causers of periodontitis, nowadays in the current consensus on the pathogenesis of periodontitis, the disease is initiated by a synergistic and dysbiotic microbial community. In this polymicrobial synergy, different members of the community, fulfilling distinct roles establishing a diseaseprovoking microbiota. One important requirement for this kind of pathogenic communities to form is the presence of certain species, like P. gingivalis (keystone pathogen) that have certain capacity to modulate the host response in ways that impair the immune surveillance and turn the balance from homeostasis to dysbiosis. This kind of microorganisms have the capacity of augment the virulence of the entire microbial community through the use of communication methods with other pathogens. Within this process, it is also important the expression of certain molecules like for example, proteolytic enzymes, proinflammatory surface structures, adhesins and other receptors. The expression of this kind of molecules by the members of such biofilms, act as community virulence factors that sustain the heterotypic proinflammatory microbial community that leads to a non-resolving and tissue destructive host response that characterize periodontitis. It should also be taken in consideration that differences intrinsic to the host response of each individual might influence the establishment and progression of the disease.
Following the reviewer suggestion, the introduction was modified in order to better explain the process. Where “In particular, Porphyromonas gingivalis has been identified as one of these keystone pathogens, able to promote dysbiosis by altering the community of commensal pathogens and leading to periodontal pathology”, page 1 lines 40-42, could be read, it now reads:
“In particular, Porphyromonas gingivalis has been identified as an example of keystone pathogen, with capacity of augmenting the virulence of the entire microbial community through specific inter-bacterial interactions (a characteristic feature of the “biofilm quorum sensing” [5,6] and the expression of certain molecules, acting as virulence factors, like proteolytic enzymes, or other pro-inflammatory molecules that will induce a dysbiosis state by modifying the biofilm
towards a pro-inflammophilic environment, thus promoting a non-resolving chronic
inflammatory host response, what is characteristic of the subgingival biofilm in periodontitis. It should also be taken in consideration that differences intrinsic to the host response of each individual might influence the establishment and progression of the disease. [7,8].”Pages 1-2, lines 40-48.
The following bibliography was also included
5 - Romero-Lastra, P.; Sanchez, M.C.; Ribeiro-Vidal, H.; Llama-Palacios, A.; Figuero, E.; Herrera, D.; Sanz, M. Comparative gene expression analysis of Porphyromonas gingivalis ATCC 33277 in planktonic and biofilms states. PLoS One 2017, 12, e0174669, doi:10.1371/journal.pone.0174669.
6 - Sanchez, M.C.; Romero-Lastra, P.; Ribeiro-Vidal, H.; Llama-Palacios, A.; Figuero, E.; Herrera, D.; Sanz, M. Comparative gene expression analysis of planktonic Porphyromonas gingivalis ATCC 33277 in the presence of a growing biofilm versus planktonic cells. BMC Microbiol 2019, 19, 58, doi:10.1186/s12866-019-1423-9.
2. -Isolated molecules are not being studied here, but rather crude extract from fish oil to form EPA and DHA, not specialized mediators such as lipoxins, resolvins etc. Please clarify (line 70-71; page 2).
Effectively, we are focusing this line of research on the antibacterial effect of each of the two fatty acids (DHA and EPA). With the positive results obtained in regard to its antimicrobial activity after 60s of exposure, we are already designing future experiments using biologically active metabolites from PUFAs, namely lipoxins, resolvins, etc.
Aiming to clarify the use of pure EPA and DHA where in the page 2 line 70 was read …. “to evaluate the antimicrobial activity of PUFAs - EPA and DHA against”... it now reads in the page 2 line 76:
…“to evaluate the antimicrobial activity of pure EPA and DHA against”…
3 -If EPA and DHA have inhibitory functions to both gram-positive and gramnegative, why are the authors continuing to study? IS that due to lack of MOI, concentration, type of bacteria? Please clarify?
The reviewer question is appropriate since we did not explain in detail the methodology used to test the antimicrobial capacity of these fatty acids. Most of the previous published investigations report the use of bacteria in planktonic state or monospecies biofilm to test the antimicrobial effect of PUFAs. In the present study we have used a multispecies subgingival biofilm model, which in our opinion represents a more valid model, since it includes representative bacteria of initial, early and late biofilm colonizers.
In order to better explain this topic, where in the page 2 lines 67-69 was read …. “However, the reported antimicrobial activity of PUFAs has been demonstrated in in vitro investigations using selected bacteria in planktonic growth, rather than on mature multispecies biofilms[37], which better resembles real conditions”…. it now reads in page 2 lines 73-75:
…“However, there are reports on the antimicrobial activity of PUFAs using planktonic bacteria or monospecies biofilms, there are no reports using validated multispecies subgingival models which better resembles real conditions.”…
The following reference was also included:
Sun, M.; Zhou, Z.; Dong, J.; Zhang, J.; Xia, Y.; Shu, R. Antibacterial and antibiofilm activities of docosahexaenoic acid (DHA) and eicosapentaenoic acid (EPA) against periodontopathic bacteria. Microb Pathog 2016, 99, 196-203, doi:10.1016/j.micpath.2016.08.025.
Methods:
4. - Time point issues. What is the justification of 70h application on the biofilms? Was there a time point or concentration dependent pilot study done before this decision? Please justify.
The rationale for selecting of 72 h biofilms was based on previous studies using this multispecies biofilm model with different antimicrobial agents. Given that each bacterial species has a different and characteristic growth rates, we have adjusted the model dynamics by monitoring the incorporation and growth rates of each species within the biofilm. Based on these studies, the 72h of biofilm growth is the time point where the biofilm reaches its most suitable maturation point, since the six bacterial strains are at an optimal concentration.
In order to better explain this topic, the following sentence was added to the page 3 lines 146-148:
“At 37°C for 72 h, time point in which the biofilm model reach maturity, containing all bacterialspecies at an optimal concentration to carry out the assay.”
The following references were added with this sentence:
37. Sanchez, M.C.; Llama-Palacios, A.; Blanc, V.; Leon, R.; Herrera, D.; Sanz, M. Structure, viability and bacterial kinetics of an in vitro biofilm model using six bacteria from the subgingival microbiota. J Periodontal Res 2011, 46, 252-260, doi:10.1111/j.1600-0765.2010.01341.x.
38. Sanchez, M.C.; Ribeiro-Vidal, H.; Esteban-Fernandez, A.; Bartolome, B.; Figuero, E.; Moreno- Arribas, M.V.; Sanz, M.; Herrera, D. Antimicrobial activity of red wine and oenological extracts against periodontal pathogens in a validated oral biofilm model. BMC Complement Altern Med 2019, 19, 145, doi:10.1186/s12906-019-2533-5.
39. Sanchez, M.C.; Fernandez, E.; Llama-Palacios, A.; Figuero, E.; Herrera, D.; Sanz, M. Response to antiseptic agents of periodontal pathogens in in vitro biofilms on titanium and zirconium surfaces. Dent Mater 2017, 33, 446-453, doi:10.1016/j.dental.2017.01.013.
5. - Concentration versus toxicity: Please provide a justification or reference. High concentration of most molecules will cause toxic effects to cells and bacterial cells. We need to understand how the authors selected the concentration?
In fact, before the start of experiments using the subgingival biofilm model, we assessed whether the tested PUFAs had any antimicrobial effect on the bacterial strains included in the model, testing each fatty acid independently with each bacterial strain in planktonic state. These data were not included in the manuscript to avoid overloading the paper with preliminar data, but in light of the query from the reviewer we have included in the revised version new sections in material and methods and results. Once we had the results from the planktonic tests, 100μM concentration was selected for the biofilm assays.
The following text in the 2.1 section, page 2 lines 75-78 was deleted “both at a concentration of 100 μM, based on the Minimum Bactericidal Concentrations (MBC) and the Minimum Inhibitory Concentrations (MIC) of EPA and DHA for the six bacterial species used in the biofilm model.”.
A new 2.3 section, page 3 lines 112-134, was created in the manuscript with the following title “Antibacterial effect of EPA and DHA against planktonic bacteria”. In this new section, the following text can be read:
“ For determining which concentration of each of EPA and DHA were appropriate for the biofilm model assays, we undertook independent previous microtiter plate-based antibacterial assays for each of the studied fatty acids. In brief, pure cultures of the 6 selected bacterial strains were grown anaerobically in a protein rich medium containing brain‐heart infusion (BHI) (Becton, Dickinson and Company, Franklin Lakes, NJ, USA) supplemented with 2.5 g L−1 mucin (Oxoid), 1.0 g L−1 yeast extract (Oxoid), 0.1 g L−1 cysteine (Sigma), 2.0 g L−1 sodium bicarbonate (Merck), 5.0 mg L−1 hemin (Sigma), 1.0 mg L−1 menadione (Merck), and 0.25% (v/v) glutamic acid (Sigma). At mid‐exponential phase of bacterial growth (measured by spectrophotometry), bacteria were placed on a 96‐wells microtitre plates adding 200μL of a mixture of each bacteria inoculum at a
final concentration of 106 colony forming units (CFUs) mL−1, and EPA or DHA for a final concentration of 12.5, 25, 50, 100 and 200μM. Plates had a set of controls: Phosphate buffered saline (PBS) was used as negative control; ethanol controls (adjusted to match the ethanol concentration present in each of the fatty acids (EtOH)); positive control (bacteria without any treatment). A measurement (optical density, O.D.595) as t = 0 absorbance was taken in a microtitre plate reader (Optic Ivymen System 2100‐C; I.C.T.; La Rioja, Spain). The microplates were incubated for 48 h at 37 °C under anaerobic conditions, and absorbance was measured each
2 h, in order to determine the bacterial growth until reaching a stationary growth phase. MIC (minimum inhibitory concentration) and MBC (minimum bactericidal concentration) values were calculated and confirmed by microbial plate counting on blood agar media. Accordingly, the lowest concentration of the DHA or EPA showing growth inhibition was considered as the MIC, whereas the lowest concentration of the DHA or EPA that showed zero growth in blood agar plates, after spot inoculation and incubation for 72 h, was recorded as the MBC. All experiments were performed in triplicate with appropriate controls.”
A new results 3.1 section, page 5 lines 239-247, was created in the manuscript with the following title “Antibacterial effect of EPA and DHA against planktonic bacteria”. In this new section, the following text can be read:
“MICs and MBCs values against the six bacterial strains selected in planktonic state were determined for each of the fatty acids. In the case of DHA, the MICs were 50 μM for S. oralis, A. naeslundii and V. parvula; 100 μM for F. nucleatum; and 25 μM for P. gingivalis and A. actinomycetemcomitans. The MBCs were 100 μM for all the six bacterial species.
In the case of EPA, the MICs were 50 μM for S. oralis and V. parvula and; 100 μM for A. naeslundii and F. nucleatum; and 25 μM for P. gingivalis and A. actinomycetemcomitans. The MBCs were 100 μM for all the six bacterial species. Based on these results, the 100 μM concentration was selected in the biofilm experiments with the two fatty acids.”
Results:
6. - Excessive tables are confusing and comparisons are not appropriate. Please provide a histogram or heatmap comparing DHA vs EPA and controls side-by-side.
The two PUFAs studies in this study were evaluated in independent experiments and for that reason one of the limitations of this work is that we were not able to compare the differences between them. Although we can suppose that there are differences, it is methodologically incorrect to compare them within this study design (it’s a limitation of the study). That is the reason why there are only comparisons between the four groups (PUFA, Ethanol, PBS or CHX).
The reviewer is totally correct when says that the tables are confusing, but we estimate that it is important to report the differences not only between the PUFAs and their controls, but also the differences between controls.
Although we tried, the fact that quantitative values are expressed in CFU/ml with high SD, due to the nature of this experiments, the resulting histograms could only be constructed without representing the SDs, which we thought it was not appropriate.
7. - Confocal quantification missing. Confocal microscopy provide layers of X-stack
images that are then compiled to a 2D figure presented on figure 1 and 3. the authors need to add MFI values of these images. MFI is mean or median fluorescence intensity to allow for evaluations on the quality and accuracy of the imaging. Supplementary or main data is fine.
If we had aimed to quantify the expression of a determined protein, MFI would be undoubtedly the best method. But since we are studying fully hydrated biofilms that include extracellular components of the matrix, MFI measurements will certainly be affected by these factors, which are not dependent from state of live or death of the bacterial cells, that was out outcome measurement, and therefore, we estimated that the use of MFT would not be appropriate. As in other studies reported in the literature, CM results are presented by reporting the live/death ratios calculating the areas occupied by green and red cells. For that reason, as we just aim to classify is the bacteria is red or green, and not if more or less SYTO9 or Propidium Iodide is bind to the cell we did not use the fluorescence intensity as a variable.
8. - Lack of labels in the figures. Both SEM and CLSM would benefit from having appropriate stainning labels. For example for green staining you add add a black box inside the photo and write in green the word FITC. this can help the reader.
In order not to do any change directly in the image that could lead to changes in the quality or loss part of the image viewing field, we added the labels related to the SYTO9 and IP in the label of the figures 1 and 3 in order for the reader better understand the meaning of the colors associated with this method. No staining was used in SEM but a brief explanation of the method was added to the labels of the figures 2 and 4.
The following text was added to the labels of the figures 1 and 3:
Specimens were stained with the LIVE/DEAD® BacLightTM Bacterial Viability Kit solution, containing SYTO 9 and Propidium Iodide nucleic acid stains. Cells with a compromised membrane that are considered to be dead or dying were stain red (PI), whereas cells with an intact membrane were stain green (SYTO9).
The following text was added to the labels of the figures 1 and 3:
“The samples were dried by critical points and coated with gold by sputtering.”
9. - Lack of scale bars to the imaging figures. Please add scale bars to all imaging figures.
The reviewer is totally right as scales are only included in figures 2 and 4 and. Following the reviewer suggestion, scale bars were added to the figures 1 and 3.
Discussion:
10. - Please discuss the behavior of biofilms on multiple materials, including crowns, titanium implants.
Following the reviewer suggestion, the following text was added to the Page 16 lines 485-486
“On the other hand, the findings of this research are expected to be comparable to other artificial surfaces used in dentistry as for example titanium and zirconia.”
The following reference was added with this sentence:
39. Sanchez, M.C.; Fernandez, E.; Llama-Palacios, A.; Figuero, E.; Herrera, D.; Sanz, M. Response to antiseptic agents of periodontal pathogens in in vitro biofilms on titanium and zirconium surfaces. Dent Mater 2017, 33, 446-453,
doi:10.1016/j.dental.2017.01.013.
11. - It is important to acknowledge that this is a monogenic biofilm and the artificial study is limited. Naturally polygenic, the oral biofilm would have evolutionary mechanisms of response that need to be discussed in the context of your data.
The authors totally agree with the reviewer. The following two texts were added to the discussion:
… “Although this model has a mixed population of six bacterial strains representing the initial, early and late colonizers present in subgingival biofilms, it does not have the diversity usually found in natural subgingival biofilms as these can reach 200 species. Our model also only includes one strain of each bacterial specie what can provide different results specially when regarding to eventual evolutionary mechanisms of response to the PUFAs”… in the page 16 lines 475-480.
and:
…. “Besides the presented limitations of the study, this kind of models should be regarded as a primary step in the research process identifying possible candidate molecules to be investigated in studies with higher level of scientific evidence.”…. in the page 16 lines 487-489.
Conclusion:
12. - Conclusions are well written, but need to provide a sentence linking to periodontal disease as was portraid in the introduction.
The reviewer is completely right. The following sentence was added to the conclusions:
“Further research is needed in order to evaluate the possible use of PUFAs in the chemical plaque control during the management of periodontitis.” page 16 lines 496-497
Round 2
Reviewer 1 Report
This revised manuscript addressed all the concerns that arose from the previous document. Therefore, I recommend it for publication.
